# Development of a Novel Double Antibody Sandwich ELISA for Quantitative Detection of Porcine Deltacoronavirus Antigen

**DOI:** 10.3390/v13122403

**Published:** 2021-11-30

**Authors:** Wei Wang, Jizong Li, Baochao Fan, Xuehan Zhang, Rongli Guo, Yongxiang Zhao, Junming Zhou, Jinzhu Zhou, Dongbo Sun, Bin Li

**Affiliations:** 1Institute of Veterinary Medicine, Jiangsu Academy of Agricultural Sciences, Key Laboratory of Veterinary Biological Engineering and Technology Ministry of Agriculture, Nanjing 210014, China; weiwang054@126.com (W.W.); lijizong22@sina.com (J.L.); fanbaochao.0405@163.com (B.F.); liuxuehan1996@hotmail.com (X.Z.); guorl1974@163.com (R.G.); zyx0920@163.com (Y.Z.); zhoujm0724@163.com (J.Z.); syzhoujz@163.com (J.Z.); 2Jiangsu Key Laboratory for Food Quality and Safety-State Key Laboratory Cultivation Base of Ministry of Science and Technology, Nanjing 210014, China; 3Jiangsu Co-Infection Center for Prevention and Control of Important Animal Infectious Disease and Zoonoses, Yangzhou 225009, China; 4Jiangsu Key Laboratory of Zoonoses, Yangzhou University, Yangzhou 225000, China; 5Laboratory for the Prevention and Control of Swine Infectious Diseases, College of Animal Science and Veterinary Medicine, Heilongjiang Bayi Agricultural University, Daqing 163319, China; dongbosun@126.com

**Keywords:** porcine deltacoronavirus, quantitative ELISA, antigen detection, intestinal and fecal samples, vaccine evaluation

## Abstract

Porcine deltacoronavirus (PDCoV) can cause diarrhea and dehydration in newborn piglets. Here, we developed a double antibody sandwich quantitative enzyme-linked immunosorbent assay (DAS-ELISA) for detection of PDCoV by using a specific monoclonal antibody against the PDCoV N protein and an anti-PDCoV rabbit polyclonal antibody. Using DAS-ELISA, the detection limit of recombinant PDCoV N protein and virus titer were approximately 0.5 ng/mL and 10^3.0^ TCID_50_/mL, respectively. A total of 59 intestinal and 205 fecal samples were screened for the presence of PDCoV by using DAS-ELISA and reverse transcriptase real-time PCR (RT-qPCR). The coincidence rate of the DAS-ELISA and RT-qPCR was 89.8%. DAS-ELISA had a sensitivity of 80.8% and specificity of 95.6%. More importantly, the DAS-ELISA could detect the antigen of PDCoV inactivated virus, and the viral antigen concentrations remained unchanged in the inactivated virus. These results suggest that DAS-ELISA could be used for antigen detection of clinical samples and inactivated vaccines. It is a novel method for detecting PDCoV infections and evaluating the PDCoV vaccine.

## 1. Introduction

Porcine deltacoronavirus (PDCoV) is an enveloped, single-stranded, positive-sense RNA virus that belongs to the genus Deltacoronavirus within the family *Coronaviridae* of the order *Nidovirales* [1]. PDCoV can causes acute diarrhea, vomiting, and dehydration in neonatal piglets [2,3]. PDCoV was first discovered in Hong Kong, China, in 2012 in a territory-wide molecular epidemiology study in mammals and birds [4]. Subsequently, in early 2014, the first outbreak of PDCoV-associated diarrhea was documented in swine, in Ohio. Among intestinal or fecal samples obtained from diarrhea pigs from five Ohio farms, 92.9% were found to be positive for PDCoV by RT-PCR [5], which then spread to many States in the USA [6]. PDCoV has also been documented in Thailand [7], Korea [8], Canada [9], Lao PDR [10], and Japan [11]. PDCoV RNA was first detected in domestic pigs in mainland China in 2014 [12]. The PDCoV infection has caused significant economic losses in the swine industry worldwide.

PDCoV is enveloped and pleomorphic with a diameter of 60–180 nm, excluding the projections. PDCoV has a single-stranded positive-sense RNA genome of approximately 25.4 kb in size (excluding the poly A-tail) that encodes four structural proteins, namely spike (S), envelope (E), membrane (M), and nucleocapsid (N), and four nonstructural proteins. The PDCoV genome organization and arrangement consist of a 5′ untranslated region, open reading frame 1a/1b (ORF1a/1b), S, E, M, nonstructural protein 6 (NS6), N, nonstructural protein 7 (NS7), and 3′UTR [3,13]. However, according to studies on other CoVs, the replicase polyproteins 1a (pp1a) and pp1ab are generally cleaved by virus-encoded proteases into 16 nonstructural proteins involved in viral transcription and replication [13]. These proteins are associated with immune modulation, viral pathogenesis and the development of diagnostic assays.

The epidemiological, clinical, and pathological features of PDCoV are similar to those of transmissible gastroenteritis virus (TGEV) and porcine epidemic diarrhea virus (PEDV) [14,15], leading to difficulties in differential diagnosis. Although several standard detection methods, for example, virus isolation, virus neutralization tests, and indirect immunofluorescence assay, are available for the detection of viruses, these techniques are time-consuming and not suitable for detection in large-scale samples [13,15,16]. Currently, RT-PCR [5,15] and reverse transcriptase real-time PCR (RT-qPCR) [6,17] methods for the detection of these viruses have been reported. However, these methods have some shortcomings, such as the need for expensive specialized equipment, the instability of RNA samples, and the possible contamination.

Enzyme-linked immunosorbent assay (ELISA) is a sensitive, specific, and convenient method for measuring macromolecular proteins, bacteria, and viruses [18]. The method uses stable reagents and inexpensive equipment, and the results are accurate and reproducible. In our study, we obtained monoclonal and polyclonal antibodies by immunizing mice and rabbits with purified recombinant N protein of PDCoV strain CZ2020 expressed in *Escherichia coli*. A double antibody sandwich quantitative ELISA (DAS-ELISA) was then established using a high-affinity monoclonal antibody (mAb) and horseradish peroxidase (HRP)-labeled rabbit polyclonal antibody as capture and detection antibodies, respectively. The assay demonstrated high sensitivity and specificity and could be used to detect a PDCoV infection in diarrheal samples and a PDCoV antigen in vaccine production.

## 2. Materials and Methods

### 2.1. Viruses, Cell Culture, and Preparation of rPDCoV-N Protein

PDCoV strain CZ2020 (GenBank accession number: OK546242) was isolated and maintained in our laboratory. The LLC-PK1 cell line was cultured in Dulbecco’s Modified Eagle’s Medium (DMEM; Thermo Fisher Scientific, Waltham, MA, USA) supplemented with antibiotics (100 units/mL of penicillin, 100 µg/mL of streptomycin, and 0.25 µg/mL of amphotericin B; Thermo Fisher Scientific) and 10% heat-inactivated fetal bovine serum (FBS; Tianhang, Huzhou, China). LLC-PK1 cells were maintained in DMEM containing 7.5 µg/mL trypsin and used to propagate PDCoV. When obviously cytopathic effects were observed, the infected cell cultures were freeze-thawed, and cell debris was removed by centrifugation at 4000× *g* for 10 min at 4 °C. The supernatant was collected and stored at −80 °C until used. SP2/0 cells were obtained as described previously [19] and were maintained in RPMI 1640 medium with 10% FBS.

PEDV(10^6.5^ TCID_50_/mL, cultured in Vero cells), TGEV(10^8.0^ TCID_50_/mL, cultured in ST cells), porcine rotavirus (PoRV, 10^7.5^ TCID_50_/mL, cultured in Marc145 cells), porcine reproductive and respiratory syndrome virus (PRRSV, 10^6.0^ TCID_50_/mL, cultured in Marc145 cells), classical swine fever virus (CSFV, 10^6.0^ TCID_50_/mL, cultured in ST cells), porcine circovirus type 2 (PCV2, 10^7.^^0^ TCID_50_/mL, cultured in PK15 cells), and porcine pseudorabies virus (PRV, 10^8.0^ TCID_50_/mL, cultured in ST cells) were conserved in the laboratory and used to determine the specificity of DAS-ELISA.

PDCoV-N whole gene was optimized to make it suitable for the *E. coli* expression system, then, the optimized gene was synthesized and constructed into clone vector pUC57. The plasmids were digested by *NdeI* and *XhoI*, and the fragments constructed into expression vector pET30a, named pET30a-N recombinant expression plasmid. The pET30a-N was transformed into *E. coli* strain BL21(DE3), then, the bacterial culture was inoculated into LB medium containing kanamycin (50 µg/mL). The bacterial solution was cultured at 37 °C and 250 rpm until the OD_600_ value reached 0.4–0.6. IPTG was added to the final concentration of 0.5 mM, and then, bacteria were induced for 5 h at 37 °C. After induction, the bacteria were pelleted by centrifugation at 4500× *g* for 30 min at 4 °C. The bacterial pellet was resuspended in phosphate buffer solution (PBS, pH 7.4) and then broken by sonication on ice. After high-speed centrifugation, the target protein was identified in the inclusion body by SDS-PAGE electrophoresis with the whole lysate, supernatant, and pellet of bacteria. The inclusion body was extracted and purified by Ni^2+^ affinity chromatography (HisTrap HP, Cytiva, Freiburg, Germany). The purified rPDCoV-N were separated by SDS-PAGE and transferred to PVDF membranes using a Bio-Rad Mini Trans-Blot Cell (Bio-Rad, Hercules, CA, USA). The membrane was incubated with the swine polyclonal antibody (1:2000) against PDCoV, followed by goat anti-pig serum (1:5000) conjugated to horseradish peroxidase, and the target protein was visualized by enhanced chemiluminescence (ECL).

### 2.2. Recombinant PDCoV-N Protein Immunization of Animals

The immunization protocol followed conventional subcutaneous injection with slight modification [19]. Briefly, 4 to 6-week-old female BALB/c mice were subcutaneously immunized with 25 µg purified rPDCoV-N protein emulsified with complete Freund’s adjuvant (Sigma-Aldrich, St. Louis, MO, USA). After 4 weeks, the mice had subcutaneous immunizations with 25 µg of rPDCoV-N in incomplete Freund’s adjuvant. Then, 2 weeks later, the rPDCoV-N protein was mixed with normal saline (0.5 mg/mL), and immunized mice were intraperitoneally injected with 100 µL/mouse.

Four female adult (12-week-old) rabbits were also immunized according to the mice immunization protocol. However, the dose of immunization was ten-fold higher than that used in mice. Blood samples were collected from the tail vein of mice or the auricular artery of rabbits on the seventh day after each booster immunization. The antibody titers were detected by indirect ELISA.

### 2.3. Preparation and Identification of Monoclonal Antibodies (mAbs) against PDCoV-N Protein

Three days prior to cell fusion, the mice were boosted with a 0.1 mL rPDCoV-N solution (0.5 mg/mL). The mice were then bled, serum samples were collected, and the antibody titers against rPDCoV-N were tested by indirect ELISA with rPDCoV-N as the coating antigen. The mouse with the highest antibody titer was euthanized, and its spleen was collected. The fusion of B lymphocytes with mouse myeloma cells was carried out by conventional methods. The resulting hybridoma cells were plated in 96-well plates and cultured in a HAT selection medium (DMEM containing 20% FBS, 100 mg/mL streptomycin, 100 IU/mL penicillin, 100 mM hypoxanthine, 16 mM thymidine, and 400 mM aminopterin). The antibody titers of hybridoma supernatants against the PDCoV N protein were screened by indirect ELISA. Positive clones were subcloned and rescreened. mAbs with high antibody titers were then purified from ascites using the octanoic acid/saturated ammonium sulfate precipitation method and subsequently purified by protein G-sepharose columns. Isotypes of obtained mAbs were determined by using a commercial mouse mAb isotyping kit (Zymed Laboratories, Carlsbad, CA, USA).

### 2.4. Rabbit Polyclonal Antibody Preparation

Two weeks after the final injection, the rabbit with the highest antibody titer was anesthetized by an intraperitoneal injection of 10% chloral hydrate to collect whole blood and obtain the serum. The rabbit serum antibody titer reached 1:243,000, detected by indirect ELISA with rPDCoV-N as the coating antigen. The antibody was purified from the serum by using octanoic acid-saturated ammonium sulfate precipitation and protein A-sepharose columns and was desalinated over a Sephadex G-25 column. The purified polyclonal antibody was stored at −80 °C. The antibody titers were assayed by indirect ELISA. Horseradish peroxidase (HRP, Sigma-Aldrich) was labeled to the purified rabbit polyclonal antibodies by conventional methods [20].

### 2.5. Selection of Antibody Pairs

Each prepared mAb against the PDCoV N protein was coated onto wells of a 96-well microtiter plate (Costar, Corning, New York, NY, USA). rPDCoV-N (50 ng/mL) or a positive sample (PDCoV cell culture) was used as the sandwich antigen, and the HRP-labeled rabbit polyclonal antibody was used as the detection antibody to perform DAS-ELISA for antibody pairing. As a negative control, the sandwich antigen was replaced with phosphate-buffered saline (PBS). The test results are expressed as the difference values of OD_450_ and OD_630_ (named OD_450–630_ value). The best antibody pairs were obtained according to the recorded result.

### 2.6. Establishment and Optimization of DAS-ELISA

We next selected the best combination of capture mouse mAbs and detection polyclonal antibody for PDCoV antigen-capture ELISA. Briefly, microplates were coated with each capture mAb at a concentration of 2 µg/mL at 4 °C overnight (100 µL/well). After blocking, serially diluted PDCoV-infected culture supernatants or uninfected controls were added into the wells in duplicate (100 µL/well), and the plates were placed in a dark environment at 25 °C for 45 min. After washing, HRP-labeled polyclonal antibodies were added at a concentration of 0.2 μg/mL (100 µL/well), and the plates were incubated at 25 °C for 30 min. Then, the wells were washed with PBS with 0.5% Tween-20, and a 3,3′,5,5′-Tetramethylbenzidine (TMB) solution (100 µL/well) was added. Fifteen minutes later, at 25 °C, sulfuric acid (2 M, 100 µL/well) was added to stop the reaction, and the absorbances at OD_450_ and OD_630_ were measured.

### 2.7. DAS-ELISA Positive and Negative Cut-Off Values

A total of 30 PDCoV negative fecal and intestinal samples were obtained from healthy piglets. These samples were diluted with PBS (0.01 M, pH 7.2) to obtain a 10% suspension (*v*/*v*), clarified by centrifugation at 4000× *g* for 10 min, the supernatant was treated with 1% Triton X-100 and 0.3% tri-n-butyl phosphate (TNBP) for 2 h at room temperature (RT) to inactivate the virus [21] and was then detected by the established DAS-ELISA with the determined optimal conditions. The critical value was figured out by the formula of X + 3SD (“X” represents the mean value of OD_450–630_ value of 30 negative samples, “3SD” represents three standard deviations).

### 2.8. Sensitivity and Specificity Analysis of DAS-ELISA

To evaluate the sensitivity of DAS-ELISA when detecting the PDCoV-N protein, 100 µL of 8, 4, 2, 1, 0.5, 0.25, and 0 ng/mL diluted rPDCoV-N protein standard with PBS (0.01 M, pH 7.2) were added into the microplates which were coated with the optimal mAb. Then, according to the optimization of DAS-ELISA, the samples were detected. The standard curve between OD_450–630_ value and the concentration of rPDCoV-N protein was developed, and the detection limit of PDCoV-N protein was confirmed.

To evaluate the sensitivity of DAS-ELISA when detecting the PDCoV virus, the PDCoV-ZC2020 virus was diluted with the same volume of medium and titrated by TCID_50_. The acquired 10^5.0^ TCID_50_/mL PDCoV cell virus was inactivated by treatment with 1% Triton X-100 and 0.3% TNBP for 2 h at room temperature, and was serially diluted 2- or 10-fold. Then, all samples were detected by using DAS-ELISA.

To evaluate the specificity of DAS-ELISA, suspensions of PEDV, TGEV, PoRV, PRRSV, CSFV, PCV2, and PRV were selected for testing. PDCoV-positive viral suspensions and PDCoV-negative samples from non-infected (mock) cell debris were also evaluated by DAS-ELISA.

### 2.9. Duplicability Analysis of DAS-ELISA

The duplicability test was carried out as described previously (31). Briefly, the intra-batch assay was determined by detecting 7 positive samples in microplates coated with capture mAb by DAS-ELISA in three parallel wells. These samples were also detected by DAS-ELISA in microplates coated with different batches of capture mAbs for inter-batch assay. All tests were repeated three times. Intra- and inter-assay coefficients of variation (%CV) were calculated using the following formula: %CV = (standard deviations (SD)/mean) × 100%.

### 2.10. Comparison of DAS-ELISA and RT-qPCR

A total of 59 intestinal and 205 fecal samples obtained from different pig farms were processed as described above and screened for the presence of PDCoV using DAS-ELISA and RT-qPCR. The qPCR primers were, respectively, forward primer (5′-ATCGACCACATGGCTCCAA-3′), reverse primer (5′-CAGCTCTT- GCCCATGTAGCTT-3′), and a probe (5′-FAM-CACACCAGTCGTTAAGCATGGCAA- GCT-BHQ1-3′). The specific procedure was as follows: 5 min at 95 °C, followed by 40 cycles of 10 s at 95 °C, and 30 s at 60 °C. The sensitivity, specificity, and accuracy were calculated using the following formulas: sensitivity = true positive/(true positive + false negative) × 100%; specificity = true negative/(true negative + false positive) × 100%; consistency = (true positive + true negative)/(true positive + false positive + true negative + false negative) × 100%. The agreement between RT-qPCR and DAS-ELISA techniques was measured with the kappa statistic value [22,23].

### 2.11. Preparation and Detection of Inactivated Viruse Antigens

The titer of PDCoV CZ2020 was tested and adjusted to 10^7.0^ TCID_50_/mL, then, the virus was inactivated with 0.05% (*v*/*v*) beta-propiolactone at 4 °C for 24 h, and heated in a water bath at 37 °C for 2 h. The live and inactivated viruses were detected by using DAS-ELISA.

### 2.12. Ethics Statement

The experimental protocol was previously approved by the Jiangsu Academy of Agricultural Sciences Experimental Animal Ethics Committee (NKYVET 2015-0127) and was performed in accordance with relevant guidelines and regulations.

## 3. Result

### 3.1. Recombinant PDCoV N Protein Expression in E. coli

Recombinant vectors were transformed into competent *E. coli* BL21 cells and induced with 0.5 mM IPTG at 37 °C for 5 h to express the recombinant protein. SDS-PAGE showed that there was an obvious target protein band at 46.0 kDa in the whole cell lysate by IPTG induction (Figure 1A). The recombinant protein existed in the supernatant and pellet of the cell lysate, but most of the protein was in the pellet (Figure 1B). The protein in the pellet was purified by Ni^2+^ affinity chromatography, and the purified recombinant protein was acquired (Figure 1C). The purified recombinant protein was analyzed by Western blot assay using anti-PDCoV swine polyclonal antibodies (Figure 1D), which demonstrated that the 46.0 kDa band was specifically recognized by the polyclonal antibodies.

### 3.2. Preparation and Characterization of mAbs

Antibody-positive subclones were obtained by the limited dilution method. Ten days after the test, positive clones were continuously isolated and selected by the limited dilution method for subcloning. A total of 12 hybridoma cell lines capable of secreting mAbs against the PDCoV-N protein were obtained and numbered 1#–12#. All hybridomas producing antibodies were detected by indirect ELISA with 0.5 as the cut-off value. A 96-well ELISA plate was coated with purified rPDCoV-N at 1 µg/mL. The antibody titers of 12 mAbs are shown in Table 1. Antibody subtypes were also determined using the isotype detection kit. The antibody subtypes of 12 mAbs are shown in Table 2.

### 3.3. Paired Antibody Selection

The 12 mAbs were selected for antibody pairing. The data are shown in Table 3. The positive sample OD_450–630_ values of 10 mAbs were more than 2.1 times that of negative sample OD_450–630_ values, which were judged as positive. mAb 2# had the highest OD_450–630_ value for positive samples, (rPDCoV-N protein and PDCoV-CZ2020 virus), and relative lower OD_450–630_ values for negative samples (PBS and LLC-PK1). Therefore, mAb 2# was selected for the development of DAS-ELISA. mAb 2# was prepared from hybridoma cell line 2#, which was subtype IgG1 and had a titer of >1:1,024,000.

### 3.4. Development of DAS-ELISA

mAb 2# and HRP-labeled rabbit polyclonal antibody were determined as the optimal capture and detector antibodies by indirect ELISA, respectively. Optimal reaction conditions of the antigen-capture assay were conducted by a checkerboard analysis of serial dilutions of capture and detection antibodies. The results showed that the optimal concentrations for capture by mAb 2# was 2 μg/mL and for detection by HRP-labeled polyclonal antibody was 0.2 µg/mL. Thirty PDCoV-negative fecal samples were used to determine the cut-off value. The mean (X) was 0.057, and the standard deviation (SD) was 0.029, thus, the critical value (X + 3SD) was 0.144, which was used to define negativity and positivity.

### 3.5. Specificity of DAS-ELISA

The specificity of DAS-ELISA was assessed by testing culture supernatants of seven other viruses: PEDV, TGEV, PoRV, PRRSV, CSFV, PCV2, and PRV. As shown in Table 4, only the PDCoV-infected cell culture presented a positive signal, and no positive results or cross-reactivity was observed for the other viruses. These results indicated that the DAS-ELISA method was specific for PDCoV detection.

### 3.6. Detection Limit of DAS-ELISA

DAS-ELISA was assayed by using two-fold serially diluted standard rPDCoV-N protein with concentrations of 8, 4, 2, 1, 0.5, 0.25, and 0 ng/mL. The standard curve between the OD_450–630_ value and the concentration of the rPDCoV-N protein was obtained as follows: Y = 0.335X + 0.116, R^2^ = 0.996. The detection limit of the rPDCoV-N protein was approximately 0.5 ng/mL (Table 5). The standard curve is shown in Figure 2.

The detection limit of virus titer was also assayed using DAS-ELISA. A 10^6.0^ TCID_50_/mL PDCoV virus was two-fold serially diluted (Table 6). The value of 10^2^^.9^ TCID_50_/mL of virus supernatant was above the critical value of 0.144, indicating that the detection limit of DAS-ELISA was about 10^3.0^ TCID_50_/mL.

### 3.7. Reproducibility of DAS-ELISA

The results of duplicability testing are shown in Table 7 and Table 8. The %CV of intra- and inter-batch duplicability tests was 0.67–7.03% and 1.43–5.16%, respectively. Variation coefficient less than 10% represented good repeatability. These results indicated that the developed DAS-ELISA was adequate for PDCoV detection.

### 3.8. Field Sample Detection

A total of 59 intestinal and 205 fecal samples were screened for the presence of PDCoV using DAS-ELISA and RT-qPCR (Table 9). DAS-ELISA was found to have 80.8% sensitivity ((73 + 11)/(87 + 17)) and 95.6% specificity ((113 + 40)/(118 + 42)) relative to RT-qPCR. The consistency of these two detection methods was ((73 + 11 + 113 + 40)/(205 + 59)) = 89.8%. In addition, the kappa values were 0.827, which is considered substantial agreement, suggesting an almost perfect agreement between the DAS-ELISA and RT-qPCR methods.

### 3.9. Detection of Inactivated Virus Antigens

The live and inactivated viruses were checked for the concentration of PDCoV-N protein by using DAS-ELISA. As shown in Table 10, the samples of 10- and 100-fold dilution multiples exceeded the test limit of DAS-ELISA, and the result was not credible. Thus, the concentration of the live virus was about 30,000 ng/mL. After inactivation, the concentration of the antigen was about 25,000 ng/mL. These results demonstrated that inactivation of beta-propiolactone had little effect on viral proteins.

## 4. Discussion

PDCoV can cause diarrhea and dehydration in sows and acute death of newborn piglets. As such, the viruses have become prevalent in pig herds worldwide, causing significant economic losses in the swine industry [13]. Early and rapid diagnosis of PDCoV is very important to prevent and control the spread of this disease. PDCoV diagnostic methods can be divided into two categories: virological and serological methods. Virological methods include the detection of virus particles (electron microscopy), detection of viral nucleic acid (various RT-PCRs and in situ hybridization), detection of viral antigen (immunofluorescence staining and immunohistochemistry), and detection of viable virus (virus isolation and swine bioassay). Serological assays can be used to detect the infection of a virus, to determine the kinetics of the antibody response to a virus infection, and to evaluate the efficacy of vaccines. The most commonly used serological assays include indirect fluorescent antibody (IFA) assay, virus neutralization (VN) test or fluorescent focus neutralization (FFN) test, enzyme linked immunosorbent assays (ELISAs), and fluorescent microsphere immunoassays (FMIA), although some of these assays have not been validated well for the detection of PDCoV antibodies.

The confirmatory finding of a PDCoV infection incorporates the detection of PDCoV RNA or antigens in the feces or intestinal substance/tissues. A diagnosis can also be made utilizing RT-PCR assays that target a conserved region of PDCoV M or N genes [2,6], IF or IHC using virus-specific mAbs or polyclonal antibodies [24,25,26], and in situ hybridization [25]. However, all of these assays are qualitative and cannot determine the exact amount of virus. Recent studies showed that fluorescent quantitative PCR could provide a sensitive method for quantifying the number of DNA templates [26,27,28], which was widely proved to be rapid, accurate, and available to detect PDCoV in laboratory facilities. qPCR determines the viral load by detecting the copies of a specific gene segment, however, it does not necessarily reflect the amount of packaged mature viral particles that might better reflect the infecting potential and risk of outbreak. Additionally, these methods have some shortcomings, for example, IF, IHC, or in situ hybridization need a long time to detect PDCoV. Furthermore, several indirect ELISAs have been developed for the detection of antibodies against PDCoV, and these include a eukaryotic expressed PDCoV S1 protein-based ELISA [29], a prokaryotic expressed PDCoV N protein-based ELISA [30], and a PDCoV whole virus-based ELISA [31]. These ELISA methods indirectly reflect the PDCoV infection by antibody detection. However, with the use of PDCoV vaccines, this method will be unscientific and not accepted. In addition, the method of detecting the PDCoV antigen by ELISA has not been reported yet.

Because of the high homology of N protein amino acid sequences from different PDCoV strains, and the high immunogenicity of this protein, the N protein seems to be a suitable antigen marker for the diagnosis of a PDCoV infection. Therefore, in this study, we used the N protein of the PDCoV strain CZ2020 as an immunogen to obtain mAbs and polyclonal antibodies. In addition, although the N protein is located inside the virus particles, we used reagents to inactivate the virus, which enabled the antibody to pass through the envelope and react with the internal N protein.

We then tested fecal and intestinal samples by using DAS-ELISA and RT-qPCR. When examining this total of 264 samples, 27 samples gave discordant results, of which 20 samples were PDCoV-positive by RT-qPCR but PDCoV-negative by DAS-ELISA. It is possible that the DAS-ELISA test may fail to detect antigens with very low viral titers in samples. Seven other samples were PDCoV-negative by RT-qPCR but PDCoV-positive by DAS-ELISA. This disagreement might be due to the presence of PCR inhibitors and nucleic acid-degrading substances in the samples, and they were retained in extracted nucleic acids, thus affecting the accuracy of qPCR. These detection errors were also observed by Sozzi et al. [32] and Fan et al. [18]. Overall, the kappa values of these two different methods were 0.827, suggesting a very high consistency between the DAS-ELISA and RT-qPCR methods.

In summary, the double antibody sandwich ELISA has higher sensitivity and specificity than indirect ELISA [33,34] and can accurately quantify antigens at the protein level with easy experimental operation [35,36], which provides an accurate and sensitive method for detecting viral pathogens and could be further applied in PDCoV detection for pigs. Therefore, in this study, we first developed a DAS-ELISA, which could be used for quantitative detection of viral antigens, by using one mouse mAb and a rabbit polyclonal antibody as capture and detection antibodies, respectively. The described assay could detect up to 0.5 ng/mL of PDCoV-N protein and 10^3.0^ TCID_50_/mL virus stock. No cross-reactivity with other similar causative agents of diarrhea and important pig pathogens, such as PEDV, TGEV, PoRV, PRRSV, CSFV, PCV2, and PRV, was observed. Furthermore, the results of field sample detection revealed a positive coincidence between DAS-ELISA and RT-qPCR. This newly developed DAS-ELISA with high sensitivity and specificity could be used as an effective method for the diagnosis of a PDCoV infection in pigs. Additionally, we can monitor the content of the PDCoV antigen in industrialized vaccine production and improve production efficiency and vaccine quality by using this method.

## Figures and Tables

**Figure 1 viruses-13-02403-f001:**
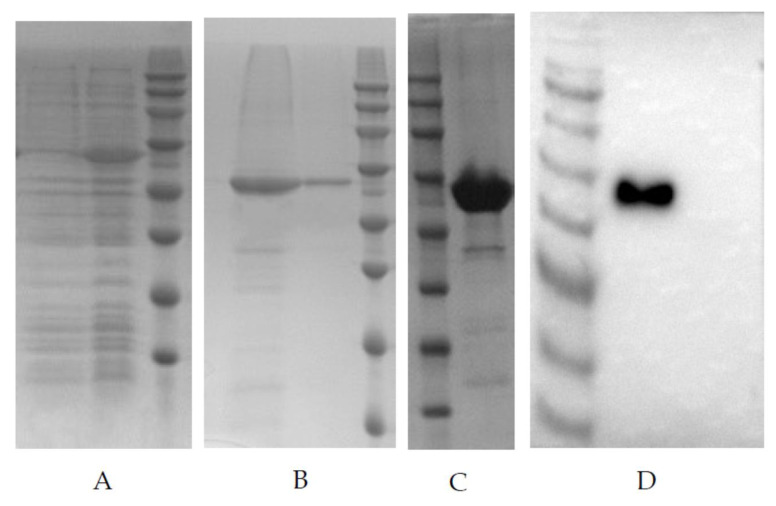
Expression and purification of the recombinant PDCoV-N protein. (**A**) SDS-PAGE of the whole lysate of transfected *E. coli* BL21(DE3); M-MW markers, 1—without induction, 2—IPTG induction. (**B**) SDS-PAGE of supernatant and pellet of the cell lysate; M-MW markers, 3—supernatant sample, 4—pellet sample. (**C**) SDS-PAGE of rPDCoV-N protein after purification; M-MW markers, 5—the purified rPDCoV-N protein. (**D**) Western blot of the purified rPDCoV-N protein; M-MW markers, 1—the purified rPDCoV-N protein, 2—the whole cell lysate without induction.

**Figure 2 viruses-13-02403-f002:**
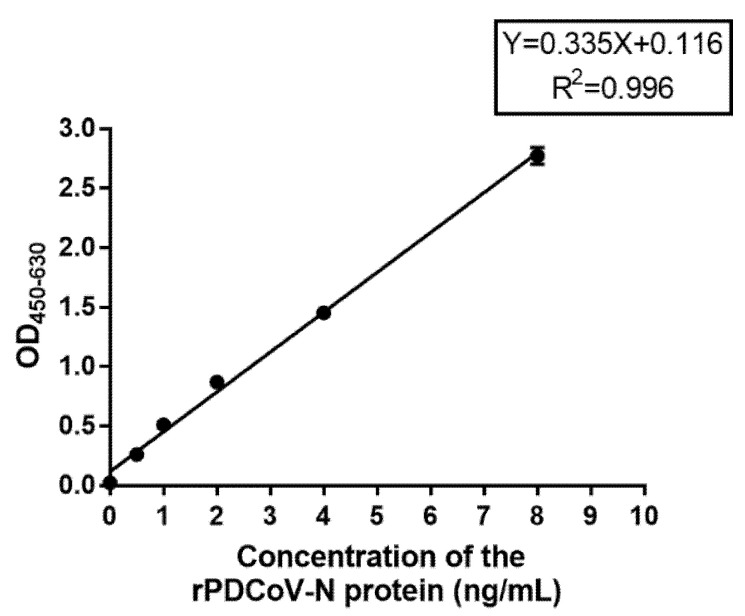
The standard curve of the DAS-ELISA. Two-fold serially diluted standard rPDCoV-N protein concentrations of 8, 4, 2, 1, 0.5, and 0 ng/mL were assayed using DAS-ELISA with two replicates per concentration. The standard curve was calculated using a linear relationship between the OD450–630 values and concentrations.

**Table 1 viruses-13-02403-t001:** The antibody titers of 12 mAbs.

Antibody Titers	Numbers of mAb
>1:1,024,000	2
1:1,024,000	3
1:25,600	6
1:9000	1

**Table 2 viruses-13-02403-t002:** The antibody subtypes of 12 mAbs.

Antibody Subtypes	Numbers of mAb
IgG1	1
IgG2a	5
IgG2b	5
IgM	1

**Table 3 viruses-13-02403-t003:** The selection of pairing antibodies.

mAbs No.	OD_450–630_ Value			
	rPDCoV-N Protein (50 ng/mL)	PBS Control	PDCoV-CZ2020 Virus	LLC-PK1 Control
1#	1.288	0.074	0.4662	0.287
2#	2.267	0.084	3.3586	0.124
3#	1.414	0.231	1.8031	0.207
4#	1.697	0.073	2.5677	0.083
5#	1.519	0.063	0.1095	0.061
6#	1.482	0.068	2.8471	0.063
7#	1.625	0.062	2.0744	0.072
8#	1.436	0.049	2.5737	0.147
9#	1.547	0.119	1.7631	0.062
10#	1.537	0.061	1.8848	0.074
11#	1.364	0.114	0.9269	0.287
12#	1.558	0.095	2.4024	0.124

**Table 4 viruses-13-02403-t004:** The results of specificity assay.

Virus	PDCoV	PEDV	TGEV	PoRV	PRRSV	PCV2	CSFV	PRV
OD_450–630_ value	1.957	0.037	0.046	0.039	0.026	0.03	0.032	0.033
PDCoV-N concentration (ng/mL)	5.50	−0.24	−0.21	−0.23	−0.27	−0.26	−0.25	−0.25

**Table 5 viruses-13-02403-t005:** The results of standard rPDCoV-N protein by using a DAS-ELISA kit.

rPDCoV-N Protein (ng/mL)	OD_450–630_ Value	
	Repeat 1	Repeat 2
8	2.725	2.823
4	1.424	1.485
2	0.884	0.857
1	0.543	0.481
0.5	0.246	0.274
0.25	0.122	0.119
0	0.025	0.024

**Table 6 viruses-13-02403-t006:** The results of PDCoV virus by using a DAS-ELISA kit.

Dilution Fold of Virus	Virus Titer (TCID_50_/mL)	OD_450–630_ Value	Concentration (ng/mL)
1:10	10^5.0^	3.314	exceed the test limit
1:20	10^4.7^	3.070	exceed the test limit
1:40	10^4.4^	2.452	6.99
1:80	10^4.1^	1.609	4.42
1:160	10^3.8^	0.949	2.41
1:320	10^3.5^	0.567	1.25
1:640	10^3.2^	0.325	0.51
1:1280	10^2.9^	0.209	0.16
1:2560	10^2.6^	0.134	−0.07
Negative control	0	0.083	−0.229

**Table 7 viruses-13-02403-t007:** The results of intra-batch duplicability test.

Assay Time	No. of PDCoV Positive Fecal Samples						
	1	2	3	4	174	187	196
First	8.27	5.48	4.03	2.48	1.97	2.40	2.08
Second	8.39	5.01	3.85	2.56	1.77	2.66	1.89
Third	8.40	4.62	3.98	2.48	1.70	2.73	1.75
X	8.35	5.04	3.95	2.51	1.81	2.60	1.91
SD	0.06	0.35	0.08	0.03	0.12	0.14	0.13
CV	0.67%	7.01%	1.95%	1.38%	6.40%	5.36%	7.03%

**Table 8 viruses-13-02403-t008:** The results of inter-batch duplicability test.

Assay Time	No. of PDCoV Positive Fecal Samples						
	1	2	3	4	174	187	196
First	8.84	5.82	4.57	2.70	1.78	2.70	1.61
Second	8.69	5.35	4.70	2.49	1.68	2.48	1.61
Third	8.54	5.36	4.29	2.64	1.63	2.39	1.47
X	8.69	5.51	4.52	2.61	1.70	2.52	1.56
SD	0.12	0.22	0.17	0.09	0.06	0.13	0.06
CV	1.43%	4.03%	3.81%	3.35%	3.65%	5.16%	4.04%

**Table 9 viruses-13-02403-t009:** Comparison of RT-qPCR and DAS-ELISA for the detection of PDCoV in intestinal and fecal samples.

		DAS-ELISA		
RT-qPCR	Fecal	Positive	Negative	Total
	Positive	73	14	87
	Negative	5	113	118
	Total	78	127	205
	Intestinal	Positive	Negative	Total
	Positive	11	6	17
	Negative	2	40	42
	Total	13	46	59

**Table 10 viruses-13-02403-t010:** Detection of live and inactivated viruses.

Dilutability	Live Virus			Inactivated Virus		
	OD_450–630_ Value	Concentration	Final Concentration	OD_450–630_ Value	Concentration	Final Concentration
		(ng/mL)	(ng/mL)		(ng/mL)	(ng/mL)
10	3.313	exceed the test limit	/	3.306	exceed the test limit	/
100	3.3	exceed the test limit	/	3.32	exceed the test limit	/
1000	2.444	6.95	20,398	2.495	7.10	20,852
2000	1.778	4.96	28,927	1.51	4.16	24,150
4000	1.161	3.12	35,862	0.907	2.36	26,808
8000	0.623	1.51	33,372	0.507	1.17	25,102

## Data Availability

All data used and presented in this study are either available in public repositories as described in the Methods section or made available in the NCBI database.

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
