# Peer review of "Development of a Novel Double Antibody Sandwich ELISA for Quantitative Detection of Porcine Deltacoronavirus Antigen"

_viruses, 2021, doi:10.3390/v13122403_

Round 1
Reviewer 1 Report
The manuscript by Wei Wang et al. describes a very novel double antibody sandwich ELISA quantitative detection for porcine deltacoronavirus. According to the authors the detection limit of recombinant PDCoV N protein and virus titer were approximately 0.5 ng/ml and 103 TCID50/ml for the DAS-ELISA. DAS-ELISA had high sensitivity and specificity for PDCoV detection. Also, the DAS-ELISA can detect the antigen of PDCoV inactivated virus. In general I find the study sound and novel but requiring some revisions.
Major concerns
*PDCoV ELISA assay could using different PDCoV antigens; like nucleocapsid (N), matrix (M), and the spike protein subunits (S1 and S2). All these antigens have been reported and are available for ELISA analysis. Please discuss the rational in this study that focus on rPDCoV-N protein.
*When assessed the specificity of DAS-ELISA, suspensions of PEDV, TGEV, PoRV, PRRSV, CSFV, PCV2 and PRV were selected for testing. How these viruses were cultured and the titer of those viruses were not mentioned in the methods.
*A total of 59 intestinal and 205 fecal samples were used for this study, how the samples were collected and stored, and were these farms exposed to PDCoV ?
*The results of this study were focused on comparison of DAS- ELISA and RT-qPCR, please include RT-qPCR protocol and primers sequencings in the material and methods.
*In this study authors examined total of 264 samples, 27 samples gave discordant results, of which 20 samples were PDCoV-positive by RT-qPCR but PDCoV-negative by DAS-ELISA. And other 7 samples were PDCoV-negative by RT-qPCR but PDCoV-positive by DAS-ELISA. What are those samples, fecal or intestinal samples, will the ways of collecting samples affect the results?
*Line 54-56, “The epidemiological, clinical, and pathological features of PDCoV are similar to those of transmissible gastroen teritis virus (TGEV) and porcine epidemic diarrhea virus (PEDV), leading to difficulties in differential diagnosis.” Please Please include references for this part.
Author Response
Responses to the comments of Reviewer #1
The manuscript by Wei Wang et al. describes a very novel double antibody sandwich ELISA quantitative detection for porcine deltacoronavirus. According to the authors the detection limit of recombinant PDCoV N protein and virus titer were approximately 0.5 ng/ml and 103 TCID50/ml for the DAS-ELISA. DAS-ELISA had high sensitivity and specificity for PDCoV detection. Also, the DAS-ELISA can detect the antigen of PDCoV inactivated virus. In general I find the study sound and novel but requiring some revisions.
Comment 1:
PDCoV ELISA assay could using different PDCoV antigens; like nucleocapsid (N), matrix (M), and the spike protein subunits (S1 and S2). All these antigens have been reported and are available for ELISA analysis. Please discuss the rational in this study that focus on rPDCoV-N protein.
Response: Thank you very much for your careful review and comment. In this study, we choose PDCoV-N protein as antigen, because of 1) the high homology of N protein amino acid sequences from different PDCoV strains, and the high immunogenicity of this protein; 2) PDCoV-N gene was more easily to be expressed in E.coli cells.
Comment 2:
When assessed the specificity of DAS-ELISA, suspensions of PEDV, TGEV, PoRV, PRRSV, CSFV, PCV2 and PRV were selected for testing. How these viruses were cultured and the titer of those viruses were not mentioned in the methods.
Response: Thanks for reviewer’s suggestion. We added these viruses information in the revised manuscript.
Comment 3:
A total of 59 intestinal and 205 fecal samples were used for this study, how the samples were collected and stored, and were these farms exposed to PDCoV ?
Response: These samples were collected from 13 farms which were occured diarrhea, and stored at -80℃. Two farms were exposed to PDCoV.
Comment 4:
The results of this study were focused on comparison of DAS-ELISA and RT-qPCR, please include RT-qPCR protocol and primers sequencings in the material and methods.
Response: Thanks for reviewer’s suggestion. We added RT-qPCR protocol and primers sequencings in the revised manuscript.
Comment 5:
In this study authors examined total of 264 samples, 27 samples gave discordant results, of which 20 samples were PDCoV-positive by RT-qPCR but PDCoV-negative by DAS-ELISA. And other 7 samples were PDCoV-negative by RT-qPCR but PDCoV-positive by DAS-ELISA. What are those samples, fecal or intestinal samples, will the ways of collecting samples affect the results?
Response: (1) “20 samples were PDCoV-positive by RT-qPCR but PDCoV-negative by DAS-ELISA”. It is possibe that the DAS-ELISA test may fail to detect antigens with very low viral titers in samples.
(2) “And other 7 samples were PDCoV-negative by RT-qPCR but PDCoV-positive by DAS-ELISA.” This disagreement might be due to the presence of PCR inhibitors and nucleic acid-degrading substances in samples, and they were retained in extracted nucleic acids, thus affecting the accuracy of qPCR.
Comment 6:
Line 54-56, “The epidemiological, clinical, and pathological features of PDCoV are similar to those of transmissible gastroen teritis virus (TGEV) and porcine epidemic diarrhea virus (PEDV), leading to difficulties in differential diagnosis.” Please include references for this part.
Response: Thanks for reviewer’s suggestion. The reference was added in the revised manuscript.
Reviewer 2 Report
The manuscript is well written and clear to the readers. The results could be useful for scientists involved in the diagnostics of PdCV. The approval of the Ethical Committee is not reported in the manuscript. Please add it (number of authorization).
Author Response
Responses to the comments of Reviewer #2
Comment 1:
The manuscript is well written and clear to the readers. The results could be useful for scientists involved in the diagnostics of PdCV. The approval of the Ethical Committee is not reported in the manuscript. Please add it (number of authorization).
Response: Thanks for reviewer’s suggestion. The approval of the Ethical Committee was added in the revised manuscript.
Reviewer 3 Report
In this manuscript, authors develop a DAS-ELISA method for quantitative detection of PDCoV antigen. The experimental design is rigorous and the writing is fluent. The experimental results have also proved to be a good method to detect the antigen of PDCoV. Very recommended to publish on Viruses. But I still have some suggestions for revising the draft.
- In line 27, if we want to detect the antigen concentration in an inactivated vaccine, SDS-PAGE and WB can be used. By the way, the major protective antigen component in a vaccine should be S antigen. I don't understand how this method can evaluate inactivated vaccines.
- In line 35 and 36, Coronaviridae and Nidovirales should be italicized.
- In line 49, ORF1a and ORF1a/1b are polyproteins that can be hydrolyzed into about 15 non-structural proteins. In addition, PDCoV is known to contain at least three accessory proteins, NS6, NS7 and NS7a, which are also non-structural proteins.
- In line 73-74 and 393, if PDCoV antigen is to be detected in vaccine production, S antigen is better than N.
- In line 125, I think authors are used the rabbit auricular artery to collect blood but not the ear marginal vein.
- In line 334, serological methods can also be used to detect viruses, not just antibodies. For example, IFA, IHC and antigen ELISA.
- In line 346, actually RT-PCR can be regarded as a semi-quantitative detection method. I don’t think DAS-ELISA can distinguish between complete viruses and incomplete viruses, and I don’t know how many mAb bind to the N protein of the same virus, so it should also be a semi-quantitative detection method.
Author Response
Responses to the comments of Reviewer #3
In this manuscript, authors develop a DAS-ELISA method for quantitative detection of PDCoV antigen. The experimental design is rigorous and the writing is fluent. The experimental results have also proved to be a good method to detect the antigen of PDCoV. Very recommended to publish on Viruses. But I still have some suggestions for revising the draft.
Comment 1:
In line 27, if we want to detect the antigen concentration in an inactivated vaccine, SDS-PAGE and WB can be used. By the way, the major protective antigen component in a vaccine should be S antigen. I don't understand how this method can evaluate inactivated vaccines.
Response: Thank you for pointing this out. SDS-PAGE and WB are regarded as a semi-quantitative method to detect antigen concentration, and these methods were time-consuming and unsensitive. In this study, the N protein of PDCoV was selected to develop a novel DAS-ELISA. This method is qualitative and can determine the amount of viral antigens up to 0.5 ng/ml and 103.0 TCID50/ml virus stock. But, the major protective antigen component in a vaccine was S protein. It is necessary to develop a novel DAS-ELISA base on S protein to detect the antigen concentration in an inactivated vaccine in future.
Comment 2:
In line 35 and 36, Coronaviridae and Nidovirales should be italicized.
Response: Thank you for pointing this out. "Coronaviridae and Nidovirales" was changed into italics.
Comment 3:
In line 49, ORF1a and ORF1a/1b are polyproteins that can be hydrolyzed into about 15 non-structural proteins. In addition, PDCoV is known to contain at least three accessory proteins, NS6, NS7 and NS7a, which are also non-structural proteins.
Response: Thank for reviewer’s reminding. We added the sentence as follows: "But according to studies on other CoVs, the replicase polyproteins 1a (pp1a) and pp1ab are generally cleaved by virus-encoded proteases into 16 non-structural proteins involved in viral transcription and replication".
Comment 4:
In line 73-74 and 393, if PDCoV antigen is to be detected in vaccine production, S antigen is better than N.
Response: Thank you very much for your comment. PDCoV S protein is major protective antigen and postulated to harbor epitopes to induce neutralizing antibodies. It is necessary to develop a novel DAS-ELISA base on S protein to detect the antigen concentration in an inactivated vaccine in future.
Comment 5:
In line 125, I think authors are used the rabbit auricular artery to collect blood but not the ear marginal vein.
Response: Thank you for pointing this out. We have replaced the "ear marginal vein" with " auricular artery".
Comment 6:
In line 334, serological methods can also be used to detect viruses, not just antibodies. For example, IFA, IHC and antigen ELISA.
Response: Thank you very much for your comment. This sentence has been revised as follows: "Serological assays can be used to detect the infection of virus, to determine kinetics of antibody response to virus infection, and to evaluate efficacy of vaccines".
Comment 7:
In line 346, actually RT-PCR can be regarded as a semi-quantitative detection method. I don’t think DAS-ELISA can distinguish between complete viruses and incomplete viruses, and I don’t know how many mAb bind to the N protein of the same virus, so it should also be a semi-quantitative detection method.
Response: Thank you very much for your comment. In this study, the standard curve was developed between OD450-OD630 value and the concentration of diluted standard rPDCoV-N protein by DAS-ELISA. By using the standard curve, the concentration of N protein was calculated, and to indicate the viral antigen load of PDCoV.